# Perceptions of Tailored Dietary Advice to Improve the Nutrient Adequacy of the Diet in French Pregnant Women

**DOI:** 10.3390/nu14010085

**Published:** 2021-12-25

**Authors:** Clélia M. Bianchi, François Mariotti, Elodie Reulet, Gaëlle Le Goff, Anne Lluch, Eric O. Verger, Jean-François Huneau, Patricia Gurviez

**Affiliations:** 1AgroParisTech, INRAE, UMR PNCA, Université Paris-Saclay, 75005 Paris, France; clelia.m.bianchi@gmail.com (C.M.B.); elodie.reulet@agroparistech.fr (E.R.); jean_francois.huneau@agroparistech.fr (J.-F.H.); 2AgroParisTech, INRAE, UMR SayFood, Université Paris-Saclay, 91700 Massy, France; gaellelegoffconfiago@gmail.com (G.L.G.); patricia.gurviez@agroparistech.fr (P.G.); 3Alimentation Science Department, Danone Nutricia Research, Centre Daniel Carasso, RD 128, 91737 Palaiseau, France; anne.lluch@danone.com; 4MoISA, University Montpellier, CIRAD, CIHEAM-IAMM, INRAE, Institut Agro, IRD, 34394 Montpellier, France; eric.verger@ird.fr

**Keywords:** pregnancy, tailored dietary counseling, mixed methods, dietary modifications, motivations, barriers and enablers

## Abstract

Tailored dietary counseling could be specifically efficient during pregnancy, a period accompanied by a rise in nutrition awareness, but little is known about the expectations of pregnant women in this regard. We studied these expectations regarding tailored dietary advice in French women during their pregnancy, as well as their motivations and the perceived barriers and enablers. In French pregnant women, we evaluated the perceptions of tailored dietary advice provided by stepwise dietary counseling based on three types of dietary changes, consisting of: (1) a modification of the amounts consumed, (2) substitutions within the food subgroups, and (3) substitutions between food subgroups. A sequential explanatory mixed-method approach was designed. Using qualitative data from a focus group study (*n* = 40), we intended to explore in depth the women’s expectations regarding dietary advice and adherence to a tailored approach. These were combined with quantitative and qualitative data from a 6-week online longitudinal study (*n* = 115), using questionnaires designed to assess the modifications of dietary habits during pregnancy and to evaluate each type of dietary change. Both studies confirmed that most women in our samples did indeed intend to institute changes regarding healthier dietary practices during pregnancy. The principal motivation behind changes to their habits was to ensure the health and well-being of both their babies and themselves. The proposal of dietary advice that is tailored to both the current diet and the specific needs of pregnant women, but that is also positive and credible, was perceived as enabling implementing healthier dietary practices during pregnancy. Regarding the implementation of the dietary changes proposed, the enablers and barriers identified differed between modifications of the amounts consumed and substitutions. The women displayed interest in all types of dietary changes. This gave relevance to combining different types of changes in order to propose dietary counseling during pregnancy. Tailored dietary counseling was identified by French pregnant women in our samples as enabling them to adopt a healthier diet. However, perceived barriers might limit the implementation of dietary changes, especially when they involved marked modifications to their usual diet.

## 1. Background

Within the framework of the life course perspective, pregnancy is a specific transitional period for women, when biological, physiological, social and emotional changes are experienced [1,2,3]. Pregnant women may therefore be keener to adopt healthier behaviors that could be pursued over time. For instance, many women stop consuming alcohol while pregnant [4,5], or quit smoking during the periconceptional period and may not return to it postpartum [4,5,6]. However, the adoption of healthier dietary behaviors is not a straightforward issue. Since women experience a rise in nutrition awareness during pregnancy [7,8,9], they seek more nutrition-related information than before [9,10,11]. They try to implement basic actions to improve their diet, such as preferring healthier options, planning their meals ahead [12], eating more fruits and vegetables, complying with dietary guidelines [13], or eating fewer unhealthy foods [13,14,15], as has been reported by many qualitative studies conducted in different developed countries [9,12,13,14,15]. However, much confusing information about nutrition-related issues is offered from various sources (social environment, healthcare providers, or mass media) and women are looking to receive credible and trustworthy dietary advice [14,16]. On the other hand, because nutrient requirements are not always satisfied among pregnant women in developed countries [17,18] and adequate maternal nutrient intakes are associated with the fetus and child development and health [19,20], it is very important to understand how appropriate dietary counseling should be provided during pregnancy.

Over the previous decade, the personalization of nutrition interventions has been the subject of increasing interest [21,22]. Although the findings of studies may have differed, depending on their design, tailored dietary interventions have been identified as being promising [22,23,24]. The adaptation of advice to individuals’ habits is known to improve the acceptability and perception of such interventions [25], so the proposal of tailored dietary advice should be more efficient than that of a “one size fits all” nature [24,25]. Moreover, computer-tailored interventions, developed using algorithms, can be dynamic and repeated in a particular individual [26] while reducing costs [27]. Our group recently developed an algorithm designed to improve the nutrient adequacy of the observed diet of French women during the periconceptional period, as measured by the PANDiet index of dietary quality, using a stepwise dietary counseling model [28]. On a theoretical basis, tailored dietary advice generated by this model proved to be more efficient in improving the nutrient adequacy of the diet of French women than generic dietary advice [28,29]. However, feedback from pregnant women is necessary to characterize the key determinants of its practical and efficient implementation, including the acceptability of the approach [30,31].

The mixed-method approach is defined as the combination of qualitative and quantitative studies to investigate one research question. This approach has become very common in the field of public health. By combining qualitative and quantitative studies, this approach enables both the exploration of a new phenomenon and evaluation of its extent [32]. In the case of pregnant women, such methods have notably been used to explore those factors related to excessive weight gain [33] or to perceptions regarding the implementation of healthy changes during this specific period [34].

The objectives of this study were to explore the expectations of French pregnant women regarding tailored dietary advice during pregnancy and to understand what the motivations, barriers, and enablers regarding the implementation of tailored dietary advice might be at this time.

## 2. Methods

### 2.1. Study Design and Ethics

A sequential explanatory mixed-method design was used to investigate the expectations of French pregnant women regarding tailored dietary advice during pregnancy, as well as their motivations and the perceived barriers and enablers regarding the implementation of such advice in their diet. Qualitative research based on focus groups was followed by quantitative research involving a 6-week online longitudinal study, with questionnaires containing a majority of closed questions and a few open-ended questions. The findings of the focus groups were used to inform the online longitudinal study, the results of which were mobilized to confirm focus group findings. In addition, the open-ended questions in the online longitudinal study provided qualitative findings that were used to confirm the quantitative results. Although the two studies involved participants from the same target populations, none of the focus group participants was involved in the online longitudinal study. Both studies were approved by the *Comité de Protection des Personnes Ile-de-France X*, a French Ethics Committee (with the identifiers “NI-2016-03-01” for the focus group study and “NI-2016-03-02” for the online longitudinal study). For the focus group study, informed consent was obtained from all participants at the time of their sessions. For the online longitudinal study, online informed consent was obtained from all participants before they answered the first questionnaire.

### 2.2. Background Information: Stepwise Dietary Counseling, Based on Three Types of Dietary Changes to Improve the Nutrient Adequacy of the Diet

Our approach was designed to improve the nutrient adequacy of the diet in each individual by providing stepwise tailored dietary changes. It was based on three types of changes and had already been described in depth elsewhere [28]. Briefly, our method required prior assessment of the diet of an individual in order to calculate their PANDiet score, aiming to evaluate the nutrient adequacy of the diet of one individual by combining the probabilities of having an adequate intake in terms of 34 nutrients. Each of the three types of dietary changes was then intended to optimize this score. Type-1 dietary changes consisted of increasing or decreasing the amount of a food item present in the observed diet and, thus, not modifying the usual food repertoire. On the other hand, the two other types of dietary changes could modify the usual food repertoire by replacing a food item present in the observed diet with either a food item from the same food subgroup (type 2), or a food item from the same food group or another food group but that was consumed at the same time, according to the French cultural meal scheme (type 3). In principle, these dietary changes were supposed to be graded according to the difficulty of implementation, from type 1 to type 3.

### 2.3. Data Collection in the Focus Group Study

The conduct and reporting of this study complied with the guidelines outlined in the consolidated criteria for reporting qualitative research (COREQ) [35]; all details are supplied in Appendix A. We conducted seven focus group sessions that involved a total of 40 pregnant French women: five sessions in Paris (Ile-de-France, France; *n* = 27) and two in Aix-en-Provence (Provence Alpes Côte d’Azur, France; *n* = 13). The criteria for eligibility required that women should be pregnant, French-speaking, had not developed gestational diabetes and were not experiencing a multiple pregnancy. Because the objective of this study was to elicit verbal interactions on diet and nutrition between pregnant women from various familial, social, and dietary backgrounds, each session involved pregnant women whose pre-pregnancy body mass index (BMI), parity and socio-occupational status all differed. The characteristics of these participants are presented according to the region of recruitment in Table 1.

Each 120-minute session was video-recorded and conducted according to standard procedures for focus groups. The first and last authors (CB and PG) designed an interview guide that included the key topics to be investigated, after a review of the literature and consultation with the project team. The guide focused on three main topics: (1) concerns, beliefs and attitudes regarding diet and nutrition during pregnancy, (2) nutrition-related information-seeking behavior, and (3) expectations with respect to tailored dietary advice during pregnancy. The data relative to topics (1) and (2) have already been presented elsewhere [9], so only the data regarding topic (3) are considered in the present study. A summary of the key questions relative to this topic in the interview guide is presented in Table 2. All the questions were open-ended. The first author (CB) moderated all the sessions. An assistant moderator attended each session to assist with note-taking, time management and video-recording, and to deal with issues such as non-verbal interactions between the participants. The participants received an incentive payment of EUR 40 after completion of the study. Data collection was ensured by the first author between March and June 2015.

### 2.4. Data Collection in the 6-Week Online Longitudinal Study

We designed a 6-week online longitudinal study to evaluate the acceptability of the dietary changes (and their associated types) identified during a simulation study as being the most efficient [28] (Figure 1).

Seventeen thousand, two hundred and forty-four women aged between 18 and 44 years, living in mainland France, and members of an online panel operated by a generalist market research company (QualiQuanti, Paris, France) were contacted by email to ask if they were pregnant and were willing to participate in the study. To register, the women had to sign the consent form electronically and answer a questionnaire on their sociodemographic profile to assess their eligibility. The non-inclusion criteria were as follows: not pregnant, more than six months pregnant (i.e., birth could occur during the study), multiple pregnancy, a specific diet linked to the dietary management of metabolic disorders or major food exclusions (e.g., vegan or gluten-free diet), no signature of the consent form (Figure 2).

Eligible participants were asked to complete a questionnaire on their usual (annual) food consumption, using a simplified Food Frequency Questionnaire (FFQ) containing 56 food subgroups. If the questionnaire was not filled in, the eligible participant was not included in the study. This questionnaire was mainly used to determine the food subgroups consumed by our participants in order to prevent us from proposing the evaluation of a dietary change that might be irrelevant to their diet. In total, 36 dietary changes (12, by type of dietary changes) were selected to build three sets of dietary changes (named A, B and C, respectively) (Figure 1). In order to propose changes that targeted different food subgroups, the 36 dietary changes had previously been selected from the 60 that were most frequently identified (20, by type of dietary changes) during a simulation study [28]. If the change concerned a food item not usually consumed by the participant, a replacement dietary change was planned. Dietary changes were assigned to one set or another so that a participant would not evaluate two dietary changes involving food items from the same subgroup at the same time.

One hundred and fifteen eligible women were finally included in the study and were randomly allocated to three groups, in which we verified that there were no significant differences in terms of age, socio-professional category, parity and self-reported nutrition awareness during pregnancy (Appendix A). The characteristics of the participants are presented by group in Table 3. From week 2 to week 5, each group evaluated the sets of dietary changes in a different order, and each participant evaluated independently the six dietary changes comprising the set (two for each type of dietary change). These evaluations were performed twice: first when the set was shown to the participant for the first time, and then after one week of reflection about whether she might incorporate each dietary change of the set into her diet. The results related to this part of the study have already been published elsewhere [28]. Each evaluation was followed by an open-ended question where participants were asked to explain their answers. In week 6, the participants were asked to fill two final questionnaires. The first was an 11-item questionnaire designed to assess salient beliefs that might impact modifications to dietary habits during pregnancy. The items were derived using concepts from the theory of planned behavior [36] and the results from the focus group study and were adapted so that they could be rapidly completed online. Three scores corresponded to the three parts of the theory of planned behavior: attitude (sum of three items related to behavioral beliefs), subjective norm (sum of two items related to normative beliefs) and perceived behavior control (sum of five items related to control beliefs) were derived from the questionnaire. The second questionnaire contained 11 items designed to evaluate the adherence of participants to the dietary counseling approach we suggested: “the potential use” (Yes—Maybe—Not), “the expected frequency of dietary advice” (seven levels, from once during pregnancy to once a day), “number of pieces of dietary advice given at one time” (from one to five or more) and to each type of dietary changes. Before answering this questionnaire, it was explained to the participants that each dietary change they evaluated belonged to a specific type and a short explanation of each type was provided. Participants who completed the study received a EUR 20 voucher. The 6-week online longitudinal study took place in June and July 2015.

### 2.5. Analysis of Qualitative Data

All focus group discussions were transcribed in full by the moderator (CB). As no previous study had been performed on the expectations of pregnant French women relative to tailored dietary advice, we did not declare any pre-determined theory before data collection. An inductive thematic approach, adapted from the grounded theory, was therefore implemented to analyze the data. This approach involves familiarization with the data, an open-coding process, and data interpretation in themes derived from identified codes [37,38]. The transcripts were double-coded independently by the same two researchers (CB and GLG) using Nvivo 11 Pro for Windows (QSR International Pty Ltd., Victoria, Australia). Discrepancies between the two researchers regarding the coded categories were identified through the software and resolved through discussion; the final codebook was then defined. The coded data were then grouped into two major themes and their sub-themes (Appendix A).

One hundred and six participants who evaluated one set of dietary changes at least once were considered for the analysis (Figure 2). Regarding 3426 dietary change evaluations, there were 2272 answers to open-ended questions collected during the online longitudinal study. Among those answers, 2219 contained information about the implementation of dietary changes. They were double-coded independently by two researchers (CB and ER) using a dedicated template, with the predetermined aim of identifying the motivations, barriers and enablers relative to the implementation of the proposed dietary changes and, then, identifying barriers and enablers for the implementation of each type of dietary changes.

### 2.6. Statistical Analysis of Quantitative Data

Percentages were used to illustrate the answers from the 80 respondents to the two final questionnaires. Four participants were excluded from the subsequent analysis on the basis of their answer to the “intention” item, which corresponded to “Rarely” or “Never” (only 2.5% of answers, respectively).

According to the general framework of the Theory of Planned Behavior, the ten elements that would impact the modification of dietary habits during pregnancy were aggregated into three scores: “Attitude” (sum of three elements), “Subjective norm” (sum of two elements) and “Perceived behavioral control” (sum of five elements). The following scoring system was applied for each element: one point for “Strongly Disagree” until five points for “Strongly agree”. Thus, values could range from three to 15 for the attitude score, from two to ten for the subjective norm score and from five to 25 for the perceived behavioral control score. The “Subjective Norm” and the “Perceived Behavioral Control” scores were calculated only for 75 participants, because of missing data regarding one element in the score of one participant. Descriptive statistics (mean, standard deviation and range) were derived for each score. Normality was checked via Kolmogorov–Smirnov tests. Logistic regression models were performed to determine whether the scores and elements that would impact the modification of dietary habits during pregnancy were associated with an intention to modify dietary habits during pregnancy (“intention”). Among potential confounders, only parity was previously found to impact the acceptability of dietary advice [28], so each model was adjusted for parity. All analyses were performed using SAS 9.1.3 (version 9.1.3, SAS Institute Inc., Cary, NC, USA).

## 3. Results

### 3.1. Modification of Dietary Habits during Pregnancy

#### 3.1.1. Qualitative Data from the Focus Group Study

In the focus group study, we were able to identify a rise in nutrition awareness among pregnant French women. They suffered from many constraints related to their pregnancy diet. They were therefore keener on adopting a healthier diet in order to regain power over their diet and to start taking care of their future baby. In order to build an environment associating the well-being and health of their baby and themselves, pregnant women asked for positive and credible information about diet [9].

#### 3.1.2. Quantitative Data from the Online Longitudinal Study

The characteristics of the rise in nutrition awareness that we perceived during focus group sessions were confirmed and quantified in the online longitudinal study. Almost a quarter of respondents (23.8%) to the final questionnaire had always intended to modify their dietary habits since the start of their pregnancy and only 5.0% had never or rarely intended to do so. The mean score for attitude was high (13.5 ± 1.4 points, i.e., 90% of the maximum), whereas the mean score for the subjective norm was more temperate (7.2 ± 1.2 points, i.e., 72% of the maximum) and the mean score for perceived behavioral control was quite low (14.5 ± 4.5 points, i.e., 58% of the maximum). Details of the items and answers are presented in Appendix A. In the logistic model between intention and the three scores, adjusted for parity, only the “attitude score” was associated with “intention” (β = −0.48; *p* < 0.05).

### 3.2. A Dietary Advice Tool for Use during Pregnancy

#### 3.2.1. Qualitative Data from the Focus Group Study

Participants in the focus groups identified features of the ideal dietary advice tool they would hope to use during their pregnancy. Personalization was the key feature they expected from such a tool. They wanted something that was “*adapted to their dietary habits and lifestyle*” (Participant 1, Group 7) and to their food preferences and cravings. Several solutions were discussed by the participants who requested various user profiles (i.e., “*craves sweet things but wants to limit her weight gain*”, “*immunized against toxoplasmosis*”, “*suffering from acid reflux*”). Some participants wanted to benefit from feedback on their daily food consumption so as to be guided toward healthier options or to “*control their mistakes*” (Participant 3, Group 1).

In terms of content, the participants expressed the need for information on not only food restrictions but also the foods and nutrients that would be preferable during pregnancy and ideas for appropriate recipes.

They insisted on the credibility of the dietary advice tool, which would need to be designed by healthcare providers with respect to both nutrition and pregnancy and “*if possible, recognized by a public health agency*” (Participant 2, Group 3).

The best device identified by participants for that tool was an app they could refer to whenever necessary.

The dietary counseling approach we proposed was quite well received by the participants; firstly, because it was designed by nutrition professionals, and secondly, because the dietary changes were tailored to their dietary habits and would “*help them to identify their mistakes if [their diet] is not balanced and then to re-balance it*” (Participant 2, Group 1). They suggested increasing the degree of personalization by taking account of individual characteristics. A large majority of participants explained that knowledge of the rise in their own nutrient adequacy score as a result of the advice would facilitate its implementation. They were concerned about the frequency of contact because they did not wish to be swamped with dietary advice. A small number of messages containing advice, sent on a weekly basis, was the preference of most participants. Some emphasized that future implementation might be hampered because the advice was based on meals in the past.

#### 3.2.2. Quantitative Data from the Online Longitudinal Study

During this study, participants were not asked to describe the ideal dietary advice tool they would need during their pregnancy. However, after being exposed to dietary advice given using our method for four weeks, 52.5% of respondents said they would use the dietary advice tool we proposed, 37.5% might use it, and 10.0% would not use it. Among the respondents that would or might use the tool (potential users), 44.2% said they would like to benefit from dietary advice on a weekly basis. When cross-referencing the frequency with the number of pieces of dietary advice preferred, 21.3% of potential users favored a combination of “Three pieces of dietary advice once a week” (among the 35 possible combinations).

### 3.3. Barriers and Enablers to the Implementation of Each Type of Dietary Advice

#### Qualitative Data from the Focus Group and from the Online Longitudinal Studies

In the focus group study, the basic principles governing the three types of dietary changes were presented to the participants. In the online longitudinal study, the participants were exposed to examples of the three types of dietary changes. The same barriers and enablers to implementing each type of dietary change were identified in both studies. The major barriers and enablers relative to each type, perceived in both studies, are presented in Table 4. As types 2 and 3 were substitutions, the barriers and enablers that they shared were combined.

Despite the explanations that were given regarding the objective of these dietary changes to globally increase the nutrient adequacy of the diet, participants in the focus group study asked for details on the nutritional benefits of each piece of advice, while participants in the online longitudinal study were more receptive to a change when they perceived its nutritional benefit.

### 3.4. Favorite Types of Dietary Changes

#### 3.4.1. Qualitative Data from the Focus Group Study

In this study, we were able to determine that women who did not wish to make major changes to their diet might favor dietary changes from type 1 (i.e., they would not modify the food they usually eat), whereas women who would not be upset by certain modifications to their diet might favor the substitutions proposed under dietary changes from types 2 and 3. However, in several groups, the idea emerged that they “*would prefer to have options from the three types* […] [they] *will feel trapped if they have to choose one type definitely*” (Participant 7, Group 5).

#### 3.4.2. Quantitative Data from the Online Longitudinal Study

Finally, more participants agreed or strongly agreed that dietary changes from types 1 and 2 would be easy to implement in their diet, which was partially in line with what we had previously shown for the longitudinal evaluation. Indeed, in that study, we found that the declared intention to incorporate dietary changes from type 1 or type 2 was higher than from type 3, but the declared intention to use dietary changes from type 1 was also higher than from type 2 [28].

When we offered them the possibility of selecting their favorite from the three types, a higher proportion of the participants favored dietary changes from type 2, and a smaller proportion identified those changes as those they do not prefer (Table 5).

Among the 76 respondents who intended to modify their dietary habits all the time or from time to time during pregnancy, no association was observed between the favorite types of dietary changes and an intention to modify their dietary habits during pregnancy (data not shown).

## 4. Discussion

By combining a focus-group study and a 6-week online longitudinal study, both including pregnant women, we showed that during this specific period of the life course, these French women truly intended to modify their dietary habits. Ensuring the health and well-being of their baby and themselves by adopting a healthier diet appeared to be the strongest motivation. However, as previously demonstrated, they suffered from a lack of positive, non-guilt-inducing and trustworthy nutrition-related information [9]; therefore, as presented in this study, they wished to benefit from dietary advice. The personalization of advice appeared to be critical to achieving successful implementation in the diet. Regarding the different types of dietary advice evaluated, modifications to the amounts consumed and minor substitutions (within the same subgroup) were identified as being the easiest to implement, with few major barriers, whereas major substitutions (between subgroups) were perceived as being more difficult to implement on a daily basis. In both studies, we were, however, able to identify women adhering to varied and/or all types of dietary changes, highlighting the point that the types of dietary changes promoted during the dietary counseling process might also be parameters that could be personalized by the woman herself.

The intention to adopt a healthier diet during pregnancy has been reported consistently in other quantitative [39,40,41] and qualitative [9,12,13,14,15] studies. In the quantitative study by Gardner et al., 67%, 57% and 45% of pregnant women intended to increase their fruit and vegetable consumption and reduce that of high-sugar and high-fat foods [39]. As regards fruit and vegetable consumption, pregnant women in Australia reported having moderately strong intentions to consume the recommended servings of fruits and vegetables [39]. Quantitative studies mainly focused on specific behaviors (consuming the recommended servings of fruits and vegetables, reducing fat or sugar consumption) as a proxy for a healthier diet. By contrast, in a recently published study, the intention to comply with overall nutrition recommendations was assessed in a sample of pregnant American women using the theory of planned behavior [41]. Their mean intention to comply with nutrition recommendations was also moderate. These findings are consistent with our finding that most women (71% in the online longitudinal study) reported that they intended to modify their dietary habits from time to time. Their intention to modify their dietary habits appeared to be closely linked to the strong motivation among pregnant women to ensure the good health and well-being of both their babies and themselves [9]. In our study, the women were convinced of the benefits of modifying their dietary habits for their own health and that of their babies, as well as to increase their vitamin and mineral intake. Furthermore, the intention to modify dietary habits results in increased planning and then a higher probability of actually modifying these habits [37]. The adoption of healthier dietary habits by pregnant women should therefore be facilitated. However, the relationship between intention and behavior is not straightforward and many social and psychological factors may affect the translation of intention into behavior [42,43], particularly in the specific period of pregnancy, which is accompanied by important social, psychological, behavioral and biological changes [3].

Our findings showed that tailoring dietary advice is a key element in pregnant women if they are actually going to follow it. During the qualitative study, these women clearly asked for dietary advice that could be tailored to them as far as possible (regarding diet, food preferences, weight gain, and also information on immunization against toxoplasmosis, for example). In the longitudinal study, pregnant women were more receptive when the food item to be replaced was very frequently consumed. Our approach using tailored dietary advice should therefore be more acceptable to pregnant women than generic dietary advice [21]. In the context of pregnancy, most studies were based on dietary interventions tailored to each woman’s diet, in order to comply with generic [44] or pregnancy-specific [45,46,47] dietary guidelines. In some cases, these approaches were enriched by behavior change techniques and/or theories [44,45,46]. In a randomized controlled trial including 120 pregnant Italian women, Di Carlo et al. used dietary data from pregnant women to evaluate their dietary habits and generate a personalized diet plan that complied with both personal preferences and specific gestational needs. This resulted in a reduction in gestational weight gain and the relative risk of presenting an excessive gestational weight gain in the intervention when compared to the control group [47]. A randomized controlled trial used two behavior change theories (the transtheoretical model and social cognitive theory) to tailor strategies in order to promote dietary changes (reduction in saturated fat intake and increase in dietary fiber intake) in 68 overweight or obese pregnant women in the United States. Unlike the study by Di Carlo et al., no personalized dietary plan was given to the women; personalization only related to the state of change of the women in terms of modifying the two dietary behaviors targeted [45]. Women in the intervention group increased their dietary fiber intake but the outcome did not differ significantly from the women receiving standard care [45]. Thus, the varied efficiency of these tailored dietary interventions during pregnancy depended on the intensity of counseling, the objective of the study, and the type of personalization (current diet, motivation to modify targeted dietary behavior, etc.). Evaluation of the acceptability and feasibility of these interventions is critical to improving their efficiency [31], which is why most studies employing personalization in pregnant women have been identified as pilot studies designed to identify the type of interventions that might be developed on a larger scale, in order to improve both maternal and infant outcomes.

Our findings regarding the perceived barriers and enablers to implementing each type of dietary change were consistent with those identified in the literature [40,48,49,50]. Indeed, we found that food preferences and habits, availability, prices, and pregnancy-specific features such as cravings, digestive disorders, and lack of energy appeared to be the most important barriers to overcome. More precisely, we showed that dietary changes from type 3, which involved the most marked changes from the initial diet, would be preferred by only a few women. Even though the barriers to implementing dietary changes from type 1 and type 2 were also identified by women in both the focus group and the online longitudinal study, they were less difficult to overcome because the changes implied by those types of changes, especially of type 1, were less in opposition with the food preferences and habits of pregnant women.

The findings of this mixed-method study should be useful for the development of strategies regarding the presentation of advice and selection of the most appropriate behavior-changing techniques [51] that will enable women to implement and sustain changes to their diet. For instance, modifications of the amounts consumed, involved in the context of dietary changes from type 1, could be presented alongside household measures in order to overcome the feeling that every item must be weighed. Information concerning impacts on the nutrient adequacy of each change could be given to women to increase their trust in the relevance of this dietary advice and, thus, their compliance with it. Finally, as mentioned by many women during the focus group study, the availability of several options, between types of dietary changes or between dietary changes within a type, will facilitate the implementation of dietary advice.

### Strengths and Limitations

The major strength of this study lies in its use of the mixed-method approach. In terms of the qualitative part, the focus group generated a dynamic exchange that favored interactions and the transmission of information between participants who shared one important, visible and very personal feature: their pregnancy. This enabled us to collect detailed information regarding the implementation of dietary changes and, thus, our tailored approach during pregnancy. Furthermore, in the online longitudinal study, the pregnant women were exposed for four weeks to the most frequent and efficient dietary changes identified by our algorithm. Thus, by combining the findings of the focus groups and open-ended questions, we were able to identify both the strengths and weaknesses of our tailored approach and define opportunities for improvement. Regarding the quantitative part, because the pregnant women were exposed to dietary changes for several weeks before completing the final questionnaires, they were able to assimilate the general approach and the three types of dietary changes before they were asked to state whether they might potentially use this advice during pregnancy, and what they would expect from such a tailored approach (frequency of advice, favorite type of dietary changes, etc.).

One limitation of our study was that in both the focus group and online longitudinal study, the participants were informed about the subject before they agreed to take part. We were, therefore, working with specific populations who were already aware of nutrition issues and were prepared to discuss or to become more informed about them. This may have resulted in greater adherence to our tailored dietary advice approach when compared to pregnant women in the general population. Nevertheless, in light of the knowledge that this specific period is generally accompanied by a rise in nutrition awareness [7], this may have been less important here than during another period of life.

Finally, dietary behavior might change over time and is known to be influenced by socio-cultural characteristics. In this study, data were collected in 2015 in pregnant French women; thus, it might be that some nutrition policies in France had contributed to modifications in the dietary behaviors of pregnant women. However, to our knowledge, no study was published regarding this issue. Thus, it might be interesting to conduct further studies in pregnant French women to update our findings. Furthermore, it might be interesting to replicate this study in other countries with different socio-cultural contexts to know if our findings could be generalized beyond pregnant French women.

## 5. Conclusions

Proposing dietary advice that is tailored to both the current diet and the specific needs of pregnant women was perceived as enabling the implementation of healthier dietary practices during pregnancy. Dietary changes involving modifications to the amounts consumed and the substitutions of food items within the same food subgroup appeared to be the most acceptable offering by the pregnant French women we studied, so these could be grouped together and proposed as part of a computer-based tool to record dietary practices. Furthermore, the findings of this mixed-method study were key to identifying practical barriers (e.g., food preferences, cooking skills, cravings) and enablers (e.g., changes close to current dietary habits, new recipe ideas), which should be useful when selecting the most appropriate behavior-change techniques and developing strategies for the presentation of tailored advice that will ensure better implementation and the maintenance of dietary changes.

## Figures and Tables

**Figure 1 nutrients-14-00085-f001:**
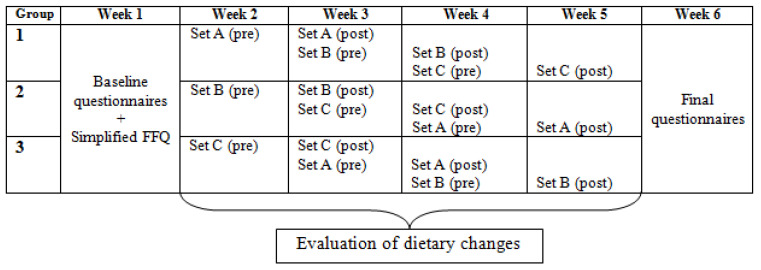
Summary of questionnaires and evaluations completed, by week and by group of participants, during the 6-week online longitudinal quantitative study. Each set of dietary changes was composed of six dietary changes (two by type). Evaluations were performed when the set was shown to the participant for the first time (pre) and then after one week of reflection about whether she might use each of the dietary changes of the set in her diet (post).

**Figure 2 nutrients-14-00085-f002:**
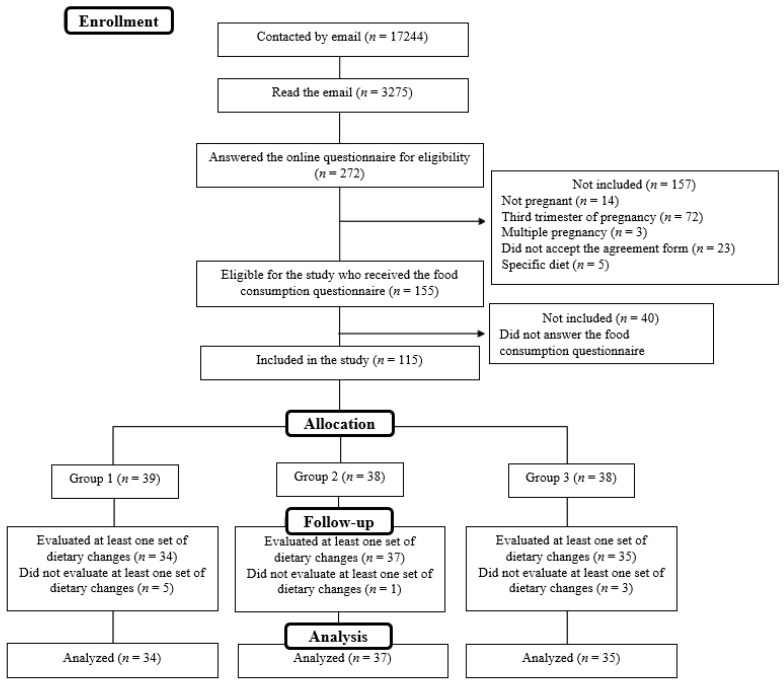
Diagram of subject flow and reasons for non-inclusions in the study of the acceptability of dietary changes during pregnancy.

**Table 1 nutrients-14-00085-t001:** Characteristics of participants by region of recruitment.

	Aix-en-Provence (*n* = 13)	Paris (*n* = 27)	Total (*n* = 40)
Age ^1^ (years)	31.9 ± 5.5	29.7 ± 3.4	30.5 ± 4.2
Pre-pregnancy BMI ^1^ (kg/m^2^)	21.8 ± 3.0	22.5 ± 3.7	22.2 ± 3.4
Trimester of pregnancy ^2^			
1st	15.4% (2)	11.1% (3)	12.5% (5)
2nd	23.1% (3)	63.0% (17)	50.0% (20)
3rd	61.5% (8)	25.9% (7)	37.5% (15)
Primiparous ^2^	46.2% (6)	51.9% (14)	50.0% (20)
Household income ^2^ (EUR per month)			
<2000	23.1% (3)	11.1% (3)	15.0% (6)
2000–4000	38.5% (5)	55.6% (15)	50.0% (20)
>4000	30.8% (4)	22.2% (6)	25.0% (10)
Did not wish to answer	7.7% (1)	11.1% (3)	10.0% (4)
Had previously followed a diet ^2^			
Never	30.8% (4)	51.9% (14)	45.0% (18)
Once	30.8% (4)	11.1% (3)	17.5% (7)
Several times	38.5% (5)	37.0% (10)	37.5% (15)

^1^ All values are mean ± SD. ^2^ All values are percentages, followed by the corresponding number of participants in brackets.

**Table 2 nutrients-14-00085-t002:** Key questions in the interview guide.

Summary of Key Questions
Have you heard about any materials (app, internet, guide book) which provide tailored dietary advice during pregnancy? If yes, do you use one of them, and why?
If you had to design the perfect dietary advice tool for pregnancy, what would it be like?
I have explained the tailored dietary counseling approach that we have designed for pregnancy; what would the pregnant women using it be like? And those not using it?
Might you adopt this tailored dietary counseling approach during your pregnancy?
Among the dietary changes I have presented, which type would be your favorite?

**Table 3 nutrients-14-00085-t003:** Characteristics of pregnant women (*n* = 115) included in the study of acceptability of dietary changes during pregnancy.

	Total (*n* = 115)	Group 1 (*n* = 39)	Group 2 (*n* = 38)	Group 3 (*n* = 38)
Age (years) ^1,4^	31.1 ± 4.2	32.0 ± 4.2	31.2 ± 4.4	30.2 ± 4.1
Months of pregnancy ^2,5^				
Less than 3	39.1% (45)	35.9% (14)	36.8% (14)	44.7% (17)
3 or 4	32.2% (37)	33.3% (13)	28.9% (11)	34.2% (13)
5 or 6	28.7% (33)	30.8% (12)	34.2% (13)	21.1% (8)
Primiparous ^2,5^	47.8% (55)	41.0% (16)	55.3% (21)	47.4% (18)
Number of people composing the household ^2,5^				
1	0.9% (1)	0.0% (0)	2.6% (1)	0.0% (0)
2	52.2% (60)	51.3% (20)	52.6% (20)	52.6% (20)
3	36.5% (42)	38.5% (15)	34.2% (13)	36.8% (14)
4	7.0% (8)	2.6% (1)	10.5% (4)	7.9% (3)
5	1.7% (2)	5.1% (2)	0.0% (0)	0.0% (0)
6 or more	1.7% (2)	2.6% (1)	0.0% (0)	2.6% (1)
Number of children ^2,5^				
0	47.8% (55)	41.0% (16)	52.6% (20)	50.0% (19)
1	40.0% (47)	46.2% (18)	34.2% (13)	39.5% (15)
2	7.8% (9)	5.1% (2)	10.5% (4)	7.9% (3)
3	1.7% (2)	5.1% (2)	0.0% (0)	0.0% (0)
4 or more	0.9% (1)	2.6% (1)	0.0% (0)	0.0% (0)
No answer	0.9% (1)	0.0% (0)	2.6% (1)	0.0% (0)
Occupation ^2^				
Farmer, Craftsperson, Storekeeper	0.9% (1)	0.0% (0)	0.0% (0)	2.6% (1)
Professional, executive	20.0% (23)	15.4% (6)	23.7% (9)	21.1% (8)
Intermediate profession	14.8% (17)	20.5% (8)	13.2% (5)	10.5% (4)
Employee	50.4% (58)	51.3% (20)	55.3% (21)	44.7% (17)
Manual worker	0.0% (0)	0.0% (0)	0.0% (0)	0.0% (0)
Student	0.9% (1)	0.0% (0)	2.6% (1)	0.0% (0)
Inactive	13.1% (15)	12.8% (5)	5.3% (2)	21.1% (8)
Socio-professional category ^2,3,5^				
High	35.7% (41)	35.9% (14)	36.8% (14)	34.2% (13)
Low	50.4% (58)	51.3% (20)	55.3% (21)	44.7% (17)
Unemployed	13.9% (16)	12.8% (5)	7.9% (3)	21.1% (8)
Urbanization of the place of residence ^2,5^				
Paris	8.7% (10)	7.7% (3)	13.2% (5)	5.3% (2)
Major city (>100,000 inhab.)	26.1% (30)	33.3% (13)	23.7% (9)	21.1% (8)
Medium-sized town (20–100,000 inhab.)	24.3% (28)	23.1% (9)	18.2% (7)	31.6% (12)
Small-sized town (2–20,000 inhab.)	25.2% (29)	17.9% (7)	26.3% (10)	31.6% (12)
Rural area	13.0% (15)	15.4% (6)	18.4% (7)	5.3% (2)
No answer	2.6% (3)	2.6% (1)	0.0% (0)	5.3% (2)
Nutrition awareness during pregnancy ^2,5^				
Much more aware	25.2% (29)	23.1% (9)	23.7% (9)	28.9% (11)
A little more aware	59.1% (68)	56.4% (22)	60.5% (23)	60.5% (23)
Not really more aware	14.8% (17)	20.5% (8)	13.2% (5)	10.5% (4)
Not more aware at all	0.9% (1)	0.0% (0)	2.6% (1)	0.0% (0)

^1^ Values are mean ± SD. ^2^ Values correspond to the percentage of participants presenting the characteristic described in the first column followed by the associated number of participants in parentheses. ^3^ Socio-professional categories were derived from occupations. “Farmer, craftsperson, storekeeper”, “Professional, executive”, and “Intermediate profession” belong to the high socio-professional category, “Employee”, “Manual worker” and “Student” belong to the low socio-professional category, and “Unemployed” belongs to the inactive socio-professional category. ^4^ No significant difference between groups for age as tested with ANOVA, *p* > 0.05. ^5^ No significant difference between groups regarding the distribution of participants, as tested with Fisher’s exact tests, *p* > 0.05.

**Table 4 nutrients-14-00085-t004:** Barriers and enablers to implementing each type of dietary change, identified by participants in both the focus group study (*n* = 40) and the online longitudinal study (*n* = 106).

	Enablers	Barriers
Type 1	No modifications to the shopping list “*it is just a matter of simplicity* […] *my shopping list is not modified*” (Participant 2, FG4) Adequacy with food preferences and cravings “*It’s all psychological what we want to eat, if I plan* [to eat] *a drumstick, I will eat a drumstick!*” (Participant 6, FG6) No profound change “*The decrease being negligible, it seems very feasible to me*” (Participant 1, OLS) Identification of mistakes to control weight gain (when decreasing amounts) “*I feel more in control of my weight gain when I am required to reduce my food intake* […] *it highlights our mistakes and we are just required to adjust the amounts*” (Participant 5, FG1)	No idea about the amount consumed (no weighing scales at home) “*My problem is that I do not weigh what I eat*” (Participant 3, FG2) Reducing amounts means being on a diet “*This type is super restrictive because it is about weight so it is frustrating*” (Participant 5, FG2) Not hungry enough to increase amounts “*I eat until satiety and I don’t want to force myself to eat more than I need*” (Participant 3, OLS) Increasing amounts means putting on weight “*I don’t want to increase my consumption so as not to put on weight*” (Participant 4, OLS)
Type 2	Easy to implement small changes “*They are both fruits, so you just have to substitute one for another*” (Participant 8, OLS) New ideas without markedly changing dietary habits “*It gives us news ideas and small alternatives*” (Participant 4, FG1) No reduction in pleasure foods “*It means we eat something nutritionally better without being too frustrated*” (Participant 1, FG7)	Seasonality “*What’s the point of replacing a seasonal product* [tomatoes] […] *with a processed one (lamb’s lettuce)?*” (Participant 6, OLS)
Type 3	New ideas which represent a means of shifting from deep-rooted habits “*It gives us ideas and breaks the routine of our usual diet*” (Participant 7, FG6)	Changes too far removed from their instantaneous cravings “*I am sorry, but I don’t want to eat fish when I crave meat*” (Participant 1, FG3) Foods not used on the same occasion “*I used mainly cheese to cook sauces or gratins so it is difficult for me to replace it with nuts*” (Participant 9, OLS) Price “*I really like fish but it is more expensive than cold cuts*” (Participant 10, OLS)
Types 2 and 3	Preference for the food items proposed “*It’s easy because we love spinach in my family*” (Participant 1, OLS)	Less variety in the diet when the proposed food item is already consumed “*Eating a huge variety of foods makes me feel really good* […]*, to give up on one food for another is nonsensical... *” (Participant 6, OLS) Strong preference for the food to be replaced “*I like pasta too much to replace it*” (Participant 2, OLS) Do not like the proposed food “*I don’t like rapeseed oil, it tastes too strong*” (Participant 5, OLS) Do not know how to cook the proposed food “ *I don’t cook fish very often*” (Participant 7, OLS)

OLS: Online Longitudinal Study. FG: Focus Group. Sentences in italics in quotation marks are the exact words (verbatim) of participants, after English translation, in both online longitudinal study and focus groups.

**Table 5 nutrients-14-00085-t005:** Final evaluation of the three types of dietary changes by pregnant women (*n* = 80): ease of implementation and ranking.

Dietary Changes from Type 1 Are Easy to Implement in the Diet	
Strongly agree	32.5% (26) ^1^
Agree	48.8% (39)
Disagree	17.5% (14)
Strongly disagree	1.3% (1)
Dietary changes from Type 2 are easy to implement in the diet	
Strongly agree	33.8% (27)
Agree	58.8% (47)
Disagree	5.0% (4)
Strongly disagree	1.3% (1)
No answer	1.3% (1)
Dietary changes from Type 3 are easy to implement in the diet	
Strongly agree	23.8% (19)
Agree	51.3% (41)
Disagree	23.8% (19)
Strongly disagree	1.3% (1)
If I had to choose, I would favor dietary changes from	
Type 1	32.5% (26)
Type 2	45.0% (36)
Type 3	22.5% (18)
If I had to choose, I would not favor dietary changes from	
Type 1	35.0% (28)
Type 2	15.0% (12)
Type 3	48.8% (39)
No answer	1.3% (1)

^1^ Percentage of respondents (number of respondents), all such values.

## Data Availability

The data presented in this study are available on request from the corresponding author. The data are not publicly available due to privacy.

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
