# Peer review of "Perceptions of Tailored Dietary Advice to Improve the Nutrient Adequacy of the Diet in French Pregnant Women"

_nutrients, 2021, doi:10.3390/nu14010085_

Round 1

Reviewer 1 Report

I found the Bianchi and col. article very interesting. The use of focus groups in combination with online survey methodology is striking and can provide great insights into the dietary behavior of pregnant women. In addition, the contributions of the focus groups can be supported by data from online surveys with a larger sample size. However, I would like to discuss some points with the authors. 

  1. Mainly, the article is very long and some parts should be more focused, otherwise, the idea of what the authors want to convey is lost. On the other hand, I suggest that you consult the guidelines of the journal, the subsections in the abstract are not necessary, section 1 is called introduction, section 2 material and methods, section 6 is not necessary.
  2. In the title, "a mixed methods study" could be deleted?
  3. Line 67-68 it is not understood what they mean by "generic nature". On the other hand, and once you get to the end of the article it is understood what mixed method is, but this concept appears in the introduction (line 79). I suggest that it be defined in the intro.
  4. The last paragraph of the intro is repeated with the first paragraph of the material and methods. I think this paragraph is more methodology than introduction.
  5. To calculate the PANDiet score, what criteria were used? does it follow the guidelines of Mediterranean, Asian, or American diets?
  6. In table 1, check the English, maybe they meant "primiparous". The variable "household income" should be defined, as well as what do they mean by the follow-up of the previous diet? in line 162 non-inclusion is "exclusion criteria"? were women with eating disorders excluded?
  7. The FFQ was answered about annually, this could be a bias, since nutritional behavior changes frequently depending on the summer season, festivities, etc. This should be included in the limitations of the study.
  8. Section 2.4. is complex and I recommend that some kind of graphical figure be included to understand it. Likewise, I believe that the flow chart and supplementary figure 4 should be part of the main text of the article.
  9. In table 3, Fisher's test is rare to report significance when any of the categories has less than 5 observations. Could we group categories and redo it? Also, in the statistical section, Fisher's test and ANOVA should be described.
  10. The results section is interesting to read and the discussion is very well argued. However, the definition of "personalization" of dietary advice (lines 419-421) seems very interesting, although the levels are not clear, what does the phenotypic and genotypic levels refer to? What implications does it have with the data of your study?
  11. Regarding the limitations of the study, apart from the aforementioned FFQ, I think it would be important to mention that the sociocultural component of nutritional behavior is essential and that these data may be influenced by the French culture and, perhaps, other barriers or facilitators are found in women from other European countries.
  12. As for the conclusions, it would be interesting if the authors could mention what specific barriers they have found in their study. 

Author Response

Authors’ response to Reviewers’ comments

REVIEWER #1

Reviewer #1: I found the Bianchi and col. article very interesting. The use of focus groups in combination with online survey methodology is striking and can provide great insights into the dietary behavior of pregnant women. In addition, the contributions of the focus groups can be supported by data from online surveys with a larger sample size. However, I would like to discuss some points with the authors. 

Author’s response: We thank the reviewer for his/her feedback on our paper. We are glad he/she thought that the combination of quantitative and qualitative methods was an appropriate methodology to provide insights into the dietary behavior of pregnant women. We took into account his/her comments and answered each of them below.

Reviewer #1: Mainly, the article is very long and some parts should be more focused, otherwise, the idea of what the authors want to convey is lost. On the other hand, I suggest that you consult the guidelines of the journal, the subsections in the abstract are not necessary, section 1 is called introduction, section 2 material and methods, section 6 is not necessary.

Author’s response: According to the reviewer’s comments we removed the subsections from the abstract.

Reviewer #1: In the title, "a mixed methods study" could be deleted?

Author’s response: We thank the reviewer for his/her comment on the title. We removed the mention “a mixed-methods study” to make it shorter.

Reviewer #1: Line 67-68 it is not understood what they mean by "generic nature". On the other hand, and once you get to the end of the article it is understood what mixed method is, but this concept appears in the introduction (line 79). I suggest that it be defined in the intro.

Author’s response: We agree with both comments mentioned by the reviewer regarding the introduction, thus we made the following modifications:

    • “a generic nature” might not be precise enough, thus we replaced it by “a “one size fits all” nature”.
    • The mixed methods concept should, indeed, be defined the first time it is mentionned in the article. Thus, we inserted the following sentence: “The mixed method approach is defined as the combination of qualitative and quantitative studies to investigate one research question” on line 72-73.

Reviewer #1: The last paragraph of the intro is repeated with the first paragraph of the material and methods. I think this paragraph is more methodology than introduction.

Author’s response: We agree with the reviewer on the redundancy between the last paragraph of the introduction and the first paragraph of the material and methods. Thus, we made this paragraph shorter in the introduction to focus only on the objectives of the study.

Reviewer #1: To calculate the PANDiet score, what criteria were used? does it follow the guidelines of Mediterranean, Asian, or American diets?

Author’s response: We thank the reviewer for his/her comment. We added brief precisions on the method used to calculate the PANDiet score and on the number of nutrients included in the calculation. We also mentionned that it aimed at evaluating the probabilities of one individual of having adequate intakes in 34 nutrients. Thus, conversely to the Mediterranean Diet Score for example, it was not based on food group intakes but on nutrient intakes.

Reviewer #1: In table 1, check the English, maybe they meant "primiparous". The variable "household income" should be defined, as well as what do they mean by the follow-up of the previous diet? in line 162 non-inclusion is "exclusion criteria"? were women with eating disorders excluded?

Author’s response: We thank the reviewer for his/her careful reading of table 1. We took into account all his/her comments to modify this table:

  • “Primiparas” was replaced by "primiparous".
  • Both variables "household income" and “had previously followed a diet” were defined in the legend below the table

Regarding the “non-inclusion” criteria in line 162 (line 170 in the revised version of the manuscript), strictly speaking the adequate term is “non-inclusion” criteria because these are the criteria used to include or not include participants, and not the criteria used to exclude participants: “Excluding” participants can only occur after they have been included. In the specific case of our study, women who presented one of the following criteria:” not pregnant, more than six months pregnant (i.e. birth could occur during the study), multiple pregnancy, specific diet linked to the dietary management of metabolic disorders or major food exclusions (e.g. vegan or gluten-free diet), no signature of the consent form”, were not included in our study sample.

Finally, we recognized that it would have been better to include “having experienced eating disorders” in the non-inclusion criteria, however, as this is a self-reported outcome, not assessed by a clinician, we thought it could be problematic.

Reviewer #1: The FFQ was answered about annually, this could be a bias, since nutritional behavior changes frequently depending on the summer season, festivities, etc. This should be included in the limitations of the study.

Author’s response: FFQ are usually answered about so as to capture the usual intake. Here, we mentioned that participants were asked to considered usual intake as the overall intake over one year when filling in the FFQ. There are of course imprecisions in the declaration of the participants, as always, but the FFQ was not used to quantitatively assess food/nutrient intakes. Indeed, we used the FFQ to determine the food subgroups consumed by our participants so as to avoid proposing a participant to evaluate a dietary change that might be irrelevant to her diet, i.e. it discuss the consumption of a food item from a food group that would never be consumed by the participant (such as asking someone who never eat fish to what extent it would be easy for her to replace a “chicken thigh” by a “grilled mackerel”). So, to our opinion, there could be no differential bias in this regard that could be due to the time when the FFQ was completed or related to the instruction to consider usual intake on an annual basis.

Reviewer #1: Section 2.4. is complex and I recommend that some kind of graphical figure be included to understand it. Likewise, I believe that the flow chart and supplementary figure 4 should be part of the main text of the article.

Author’s response: We thank the reviewer for his/her comment which helped us to make the subsection 2.4 easier to read.

Firstly, we put the figure 1 earlier, in order to directly provide details on the sequency of questionnaires during the study. Then, we also added a reference to this figure, further in subsection 2.4, when first mentioning the set of dietary changes, to make it more visual and easier to understand for the reader.

Secondly, we thank the reviewer for suggesting to include the flow chart (former Supplemental Material 2) in the main text. We made this modification and the flow chart is now referenced as “Figure 2” (line 173). The flow chart was thus removed from the supplemental material and the Supplemental Materials 3 and 4 were thus re-numbered Supplemental Materials 2 and 3 respectively.

Finally, we thank the reviewer for his/her comment on the inclusion of the supplemental material 4 (re-numbered supplemental material 3 in the revised manuscript) in the main text. Indeed, we thought it would be of interest, however we have to maintain a balance between the length of the publication and the data we included. Therefore, we made the choice to include this table in supplemental material to avoid an overload of data in a paper which was already quite long and informative. Thank you again.

Reviewer #1: In table 3, Fisher's test is rare to report significance when any of the categories has less than 5 observations. Could we group categories and redo it? Also, in the statistical section, Fisher's test and ANOVA should be described.

Author’s response: We thank the reviewer for this comment. Indeed, we should not have done these statistical comparisons, because we are testing differences in characteristics between groups that result from random allocation. There is now a consensus among statistical advisors to consider that this should not be done in randomized trials. This has been discussed many time, see for instance https://www.ncbi.nlm.nih.gov/pmc/articles/PMC4310023/ and it is now part of the official statistical guidelines. So our decision is to remove those statistical tests from Table 3. This naturally resolve the point raised by the reviewer. Thank you again.

Reviewer #1: The results section is interesting to read and the discussion is very well argued. However, the definition of "personalization" of dietary advice (lines 419-421) seems very interesting, although the levels are not clear, what does the phenotypic and genotypic levels refer to? What implications does it have with the data of your study?

Author’s response: We thank the reviewer for his/her comment. Indeed, we agreed that the concept of levels of personalization of dietary advice is really interesting but it would require to be explained in more details. More information on this concept could be found in the review from Celis-Morales et al. , entitled “Personalising nutritional guidance for more effective behaviour change.” https://pubmed.ncbi.nlm.nih.gov/25497396/

However, as mentionned by the reviewer, this concept does not have implications with the data in our study. Thus, we decided to remove the part of this sentence mentioning the levels and to keep it short. We thank again the reviewer for his/her comment, it was helpful to make this part of the discussion less confusing.

Reviewer #1: Regarding the limitations of the study, apart from the aforementioned FFQ, I think it would be important to mention that the sociocultural component of nutritional behavior is essential and that these data may be influenced by the French culture and, perhaps, other barriers or facilitators are found in women from other European countries.

Author’s response: We thank the reviewer for his/her comment. We added these points in the limitations subsection of the discussion (lines 495-501).

Reviewer #1: As for the conclusions, it would be interesting if the authors could mention what specific barriers they have found in their study. 

Author’s response: According to the suggestion of the reviewer, we added main barriers and enablers identified in our study between brackets in the conclusion (lines 510-512).

Reviewer 2 Report

Well designed, conducted and presented.

Author Response

We thank the reviewer for his/her positive feedback on our paper, we are glad that he/she thought our paper to be well designed, conducted and presented.

Reviewer 3 Report

The manuscript entitled ‘Perceptions of tailored dietary advice to improve the nutrient adequacy of the diet in French pregnant women: a mixed methods study’ presents interesting issue, however some minor corrections are needed

Abstract:

  • The abstract should be a single paragraph and should follow the style of structured abstracts, but without headings

Introduction:

  • The introduction section is well written. I really enjoyed reading this section

Material and method:

  • The data is a little bit old (Data collection was ensured by the first author between March.) – this aspect must be discussed and indicated as a one of limitations
  • Line 162 – ‘The non-inclusion criteria’ – it should be ‘the exclusion criteria’
  • Values are mean ± SD - Was the normality of distribution tested? The information about it should be added and authors should be consequent. If data have normal distribution, they should be treated as such, if not, nonparametric tests should be applied. Please specify it.

Conclusion

  • Authors should avoid to use the references in this section.

Author Response

REVIEWER #3

Reviewer #3: “The manuscript entitled ‘Perceptions of tailored dietary advice to improve the nutrient adequacy of the diet in French pregnant women: a mixed methods study’ presents interesting issue, however some minor corrections are needed

Authors’ response: We thank the reviewer for his/her feedback on our manuscript. We are glad he/she thought our paper to present interesting issue. We took his/her comments into account and modify the manuscript accordingly.

Reviewer #3:

Abstract:

  • The abstract should be a single paragraph and should follow the style of structured abstracts, but without headings

Authors’ response: According to the reviewer’s comments we removed the subsections from the abstract.

Reviewer #3:

Introduction:

  • The introduction section is well written. I really enjoyed reading this section

Authors’ response: We thank the reviewer for his/her positive feedback on the introduction.

Reviewer #3:

Material and method:

  • The data is a little bit old (Data collection was ensured by the first author between March.) – this aspect must be discussed and indicated as a one of limitations

Authors’ response: Indeed, we agree that data were collected 6 years and that it might be a limitation of our findings. We included this point in the limitation section (lines 490-497).

Reviewer #3:

  • Line 162 – ‘The non-inclusion criteria’ – it should be ‘the exclusion criteria’

Authors’ response: Regarding the “non-inclusion” criteria in line 162 (line 170 in the revised version of the manuscript), strictly speaking the adequate term is “non-inclusion” criteria because these are the criteria used to include or not include participants, and not the criteria used to exclude participants: “Excluding” participants can only occur after they have been included. In the specific case of our study, women who presented one of the following criteria:” not pregnant, more than six months pregnant (i.e. birth could occur during the study), multiple pregnancy, specific diet linked to the dietary management of metabolic disorders or major food exclusions (e.g. vegan or gluten-free diet), no signature of the consent form”, were not included in our study sample

Reviewer #3:

  • Values are mean ± SD - Was the normality of distribution tested? The information about it should be added and authors should be consequent. If data have normal distribution, they should be treated as such, if not, nonparametric tests should be applied. Please specify it.

Authors’ response: We thank the reviewer for his/her comment. We tested the normality of distribution before performing the statistical analysis, thus we added this point in the statistical analysis subsection of the Material & Methods (lines 266-267).

Reviewer #3:

Conclusion

  • Authors should avoid to use the references in this section.

Authors’ response: The references mentioned in the conclusion section were removed.

Round 2

Reviewer 1 Report

Thank you for the update manuscript. In my opinion, it has been improved.